# Structural characterization of a soil viral auxiliary metabolic gene product – a functional chitosanase

Ruonan Wu [1,8], Clyde A. Smith [2,8], Garry W. Buchko [1,3], Ian K. Blaby [4], David Paez-Espino[5], Nikos C. Kyrpides [4], Yasuo Yoshikuni [4], Jason E. McDermott [1,6], Kirsten S. Hofmockel [1], John R. Cort[1,7] & Janet K. Jansson [1] ✉

Metagenomics is unearthing the previously hidden world of soil viruses. Many soil viral sequences in metagenomes contain putative auxiliary metabolic genes (AMGs) that are not associated with viral replication. Here, we establish that AMGs on soil viruses actually produce functional, active proteins. We focus on AMGs that potentially encode chitosanase enzymes that metabolize chitin – a common carbon polymer. We express and functionally screen several chitosanase genes identified from environmental metagenomes. One expressed protein showing endo-chitosanase activity (V-Csn) is crystalized and structurally characterized at ultra-high resolution, thus representing the structure of a soil viral AMG product. This structure provides details about the active site, and together with structure models determined using AlphaFold, facilitates understanding of substrate specificity and enzyme mechanism. Our findings support the hypothesis that soil viruses contribute auxiliary functions to their hosts.

Recent metagenomic surveys have revealed a high diversity of DNA viruses across a range of soil habitats, including permafrost[1,2], thawed permafrost[3] and grasslands[4]. The majority of these viruses are bacteriophages[3,4], although several eukaryotic viruses have also been detected[1,2]. A fundamental and largely unanswered question concerns the functional roles of these viruses in the soil habitat. It is recognized that soil bacteriophages play a major role in the regulation of host dynamics[5]. Intriguingly, soil viruses may also contribute directly towards biogeochemical processes in soil through expression of genes that potentially encode functions not directly required for viral reproduction. Genes that correspond to non-essential viral functions are referred to as auxiliary metabolic genes (AMGs). Potential functions encoded by AMGs include carbon metabolism, sporulation and

energy generation[6,7]. However, only one soil viral AMG has been expressed and functionally characterized to date[3] and there are no existing crystal structures for soil viral AMGs.

The study of AMGs in soil viruses lags behind that of marine environments, due to the high diversity and complexity of soil habitats that has confounded viral discovery. In marine ecosystems, viral AMGs that encode photosynthetic proteins have been extensively studied[8,9]. For example, some marine viruses have been shown to express *psbA* genes, with transcripts increasing during infection of the hosts[10,11]. At the protein level, the structure of a plastocyanin protein encoded by a cyanobacterial phage (cyanophage) was modeled based on a related reference structure from *Synechococcus* sp. PCC7942[8]. By comparison to the similarly modeled structure from the host plastocyanin, it was

[1]Earth and Biological Sciences Directorate, Pacific Northwest National Laboratory, Richland, WA, USA. [2]Stanford Synchrotron Radiation Light source, Stanford University, Menlo Park, CA, USA. [3]School of Molecular Biosciences, Washington State University, Pullman, WA, USA. [4]US Department of Energy Joint Genome Institute, Lawrence Berkeley National Laboratory, Berkeley, CA, USA. [5]Mammoth Biosciences, Brisbane, CA, USA. [6]Department of Molecular Microbiology and Immunology, Oregon Health & Science University, Portland, OR, USA. [7]Institute of Biological Chemistry, Washington State University, Pullman, WA, USA. [8]These authors contributed equally: Ruonan Wu, Clyde A. Smith. ✉e-mail: janet.jansson@pnnl.gov

possible to predict cyanophage-specific modifications to the structure and electrostatic potential of the cyanophage-encoded plastocyanin. In addition, the structure of a viral rhodopsin has recently been characterized at 1.4 Å resolution. The structure revealed that the viral rhodopsins are unique light-gated channels that have a predicted role in supporting photosynthesis of algae[12]. These recent discoveries in marine viruses highlight the ecological importance of AMGs that potentially maximize the fitness of phages and hosts in the environment.

The first AMGs that were described in soil viruses were genes encoding enzymes for degradation of various organic compounds. For example, 14 glycoside hydrolase genes were detected in metagenomes from thawed permafrost. One of these, a viral gene encoding a glycosyl hydrolase group 5 (GH5) enzyme, was cloned, expressed and found to represent a functional endomannanase[3]. The vast majority of predicted soil viral AMGs have been assigned potential functions solely based on their sequence similarities to annotated genes in microbial genomic databases[4,5]. This approach is however limited in its ability to determine if the AMG is actually expressed and if the protein is functional.

AMGs have been found on soil viruses that potentially encode genes involved in the decomposition of chitin[4,13], the second most abundant carbon polymer on the planet after cellulose[14]. In open oceans, Picocyanobacteria utilize chitin that is mainly produced by arthropods as a key nutrient source[15]. Chitin can also accumulate in soils because it is a component of fungal cell walls and insect exoskeletons. Following deacetylation of the chitin polymer into chitosan by chitin deacetylases, chitosanases cleave chitosan into smaller subunits that can be further degraded, thereby providing carbon and nitrogen sources for other members of the microbiota[15] Chitosanase genes have previously been annotated in the genome of a giant virus, *Chlorovirus*, that infects green microalgae in terrestrial waters[16] and the detected viral chitosanases were characterized as belonging to the glycosyl hydrolase group 46 (GH46). Chitosanase-like AMGs carried on soil viruses primarily fall into another group previously categorized as GH75 fungal chitosanases (pfam07335) that cleave beta-1,4-chitosans with endo-splitting activity.

Here we characterize and validate the function and structure of a soil viral chitosanase AMG product. We confirm that the chitosanase is indeed functional by cloning and expressing the gene and conducting activity assays. We obtain an ultra-high resolution crystal structure of the enzyme, that provides details of the potential active site for the GH75 family of chitosanases.

## Results and discussion
### Phylogeny of viral chitosanases
Viral contigs that carried GH75 chitosanase AMGs were retrieved from the Integrated Microbial Genomes and Virome (IMG/VR) database (v3.0). A total of 142 qualified GH75 chitosanase-like AMGs were identified from viral contigs with lengths ranging from 8 to 202 kb. The majority of the sequences were from bacteriophages with unclassified taxonomy. Two of the viral contigs were high-quality complete and circularized genomes (Supplementary Table 1).

A protein tree was constructed from the sequence data to delineate the relatedness of viral chitosanases to other microbial chitosanases deposited in public databases. The viral chitosanases were phylogenetically distinct from archaeal, fungal and bacterial GH75 chitosanases (Fig. 1). The GH75 chitosanases mainly clustered into separate clades according to their taxonomic assignments, except for the archaeal chitosanases, which could be due to a deficiency of archaeal representatives in the current NCBI database. The majority of the detected viral chitosanases formed tight clades that were related to bacterial chitosanases (Fig. 1). The phylogenetic placement of the viral chitosanases suggests that they originated from bacteria via genomic exchange and were modified into virus-specific versions during

genetic drift and diversification processes[17]. This hypothesis is further supported by the finding that the viral contigs that contained GH75 chitosanases were classified as bacteriophages (Supplementary Table 1).

A multiple sequence alignment was constructed to determine whether the viral chitosanases contained four conserved residues residing in a presumed active site for bacterial and fungal GH75 chitosanase sequences[18,19] (Supplementary Data 1). The GH75 viral chitosanases generally retain the same four key residues for the predicted active sites, D-D-D-E, although some of the sequences had substitutions. The majority of the GH75 chitosanases have aspartate at the first of the four positions, with a few instances of substitutions of cysteine or asparagine (innermost ring in Fig. 1). As substitutions in the key residues may affect the predicted function, we further divided these viral chitosanases into sub-groups, as it appears that incidences of active site residue variants described above tend to cluster together. The identified viral chitosanases were clustered into three major clades ('Clade 1', 'Clade 2' and 'Clade 3' in Fig. 1). The substitutions of cysteine (relative frequency 14.79%, Fig. 1) and asparagine (relative frequency 7.75%, Fig. 1) at the first position ('D34') were only observed in Clade 1 and 3 viral chitosanases, respectively. Clade 2 and Group 2 viral chitosanases contained the same residues as the majority of the bacterial and fungal chitosanases. Viral chitosanases in Clade 1 with and without cysteine substitutions were named Group 1.1 and Group 1.2, respectively, and those in Clade 3 with and without asparagine substitutions were named Group 3.1 and Group 3.2, respectively. Some of the viral chitosanases grouped according to sample origin with soil viruses grouping separately from aquatic viruses. The soil specific sequences were primarily in Group 1.2 and Clade 3.

### AlphaFold predictions
Representatives from the different Clades (Fig. 1) were selected for structural prediction using the recently introduced artificial intelligence-based protein structure prediction software AlphaFold[20]. The structures of several bacterial and fungal GH75 chitosanases that had been previously reported and characterized in the literature were also predicted. Two characteristic regions of sequence shared by all GH75 members formed a glycosyl hydrolase domain fold seen in other families, for example GH45. These shared regions bracket an uncharacterized domain, a variable region containing dissimilar insertions of varying lengths (Supplementary Fig. 1). In several predicted structures of viral AMG products this insertion folds as an uncharacterized domain of about 70 amino acids that forms one side of a prominent cleft in the middle of the entire structure. Some bacterial members of GH75 appear to contain a homologous insertion (Supplementary Fig. 1). Other viral chitosanase-like AMG sequences and all of the predicted bacterial and fungal sequences lack this longer insertion and do not appear to form a substantial domain. Insofar as these sequences have non-homologous regions of sequence, these structure predictions may be helpful in comparative analysis of the different groups.

### Identification of a chitosanase-like AMG with chitosanase activity
The DNA coding sequences for 10 of the 142 GH75 chitosanase-like AMGs were codon-optimized for recombinant expression in *Escherichia coli* (selected sequences are indicated with asterisks in Fig. 1). Expression was observed for nine out of the 10 proteins, but protein was observed primarily in the insoluble fractions in all but two of the initial targets. For the two other targets, enough protein was expressed and purified to assay for endo-chitosanase activity. Only one expressed protein, hereafter called V-Csn (for Viral Chitosanase), showed activity and this activity was maximum near pH 5 (Fig. 2a). The corresponding sequence originated from Group 3.1 and is indicated with two asterisks in Fig. 1, with the predicted D-D-D-E active site residues. This specific sequence originated from a forest soil metagenome (Supplementary

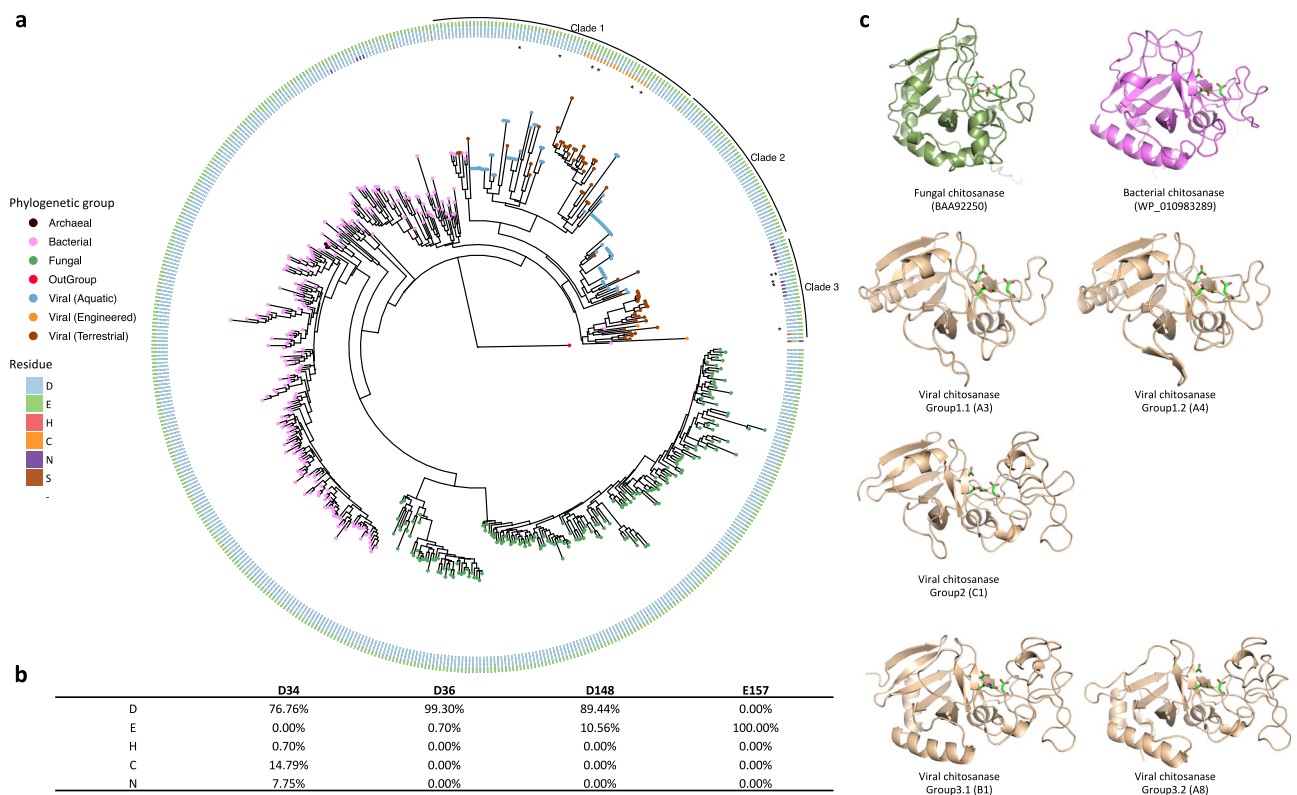

**Fig. 1 | GH75 chitosanases detected across domains of life. a** The phylogenetic tree with viral, fungal (green), bacterial (pink) and archaeal (black) chitosanases is rooted by a bacteriophage lysozyme (YP_006987285.1, red node). The tree leaves representing viral chitosanases are colored by habitat: aquatic (blue), engineered (orange), terrestrial (brown). The four key residues in the proposed active site are color coded and shown in the circular rings sequentially with the residue closest to the N-terminal position in the innermost ring. The viral chitosanases selected for enzymatic function validation are highlighted with asterisks. The viral chitosanase used for crystallization is labeled with two asterisks. **b** The relative frequencies of each residue at the four conserved sites were calculated and shown in the table. **c** The protein structures of representative chitosanases were predicted by AlphaFold[20] and are colored according to phylogenetic groups: bacterial (pink), fungal (green), viral (brown).

Table 1). The virus that carried V-Csn was predicted to be a Proteobacteria phage (Supplementary Table 1). Endo-chitosanase activity for V-Csn was further corroborated by two single-residue substitutions at positions proposed to be part of the chitosanase active site. It was previously postulated, based on biochemical, kinetic, and mutational studies on the fungal GH75 chitosanases from *Aspergillus fumigatus*[21] and *Fusarium solani*[19], that two residues (D160 and E169 in *A. fumigatus*, and D175 and E188 in *F. solani*) were essential catalytic residues. Sequence analysis of a bacterial GH75 chitosanase from *Streptomyces avermitilis* showed that these residues were also conserved in this enzyme[18]. Based upon a partial alignment of the V-Csn, *A. fumigatus*, *F. solani*, and *S. avermitilis* sequences (Fig. 2b), the corresponding residues in V-Csn are D148 and E157. Two V-Csn constructs harboring either a D148N substitution or a E157Q substitution were generated, and the activities of the respective mutant enzymes measured. Activity was reduced five-fold for D148N and almost eliminated for E157Q (Fig. 2c). Both constructs eluted with gel-filtration chromatography retention times identical to native V-Csn, and circular dichroism spectra indicated both constructs were folded. Crystallization trials on V-Csn and the two mutant enzymes were subsequently undertaken.

**High resolution X-ray structure of V-Csn**

The V-Csn structure was solved in two crystal forms; apo1 containing a single molecule in the asymmetric unit, and apo2 containing a dimer in the asymmetric unit. The apo1 structure was solved by single anomalous diffraction (SAD) methods using the signal from bromide ions soaked into crystals of the apo1 form. The structure was automatically built using phenix.autobuild[22] and completed with COOT (v0.9.8.2)[23]. The bromide anomalous signal extended to approximately 1.5 Å

resolution and the structure was initially refined into this data. Refinement was completed with phenix.refine[24] against a high resolution apo1 data set extending to 0.89 Å resolution. The final model comprised 1811 protein atoms in a single chain, 398 water molecules, three glycerol and a sulfate anion. The final $R_{work}$ and $R_{free}$ were 0.1198 and 0.1307 for 174,427 total reflections. The apo2 crystal form was solved by molecular replacement using the refined apo1 structure as the search model, and refined with phenix.refine.

A single V-Csn molecule was located in the apo1 asymmetric unit. The structure consisted of two non-contiguous structural domains (Fig. 2d). The N-terminal part of Domain-1 (residues 1-36) is folded first and the polypeptide then folds the entire Domain-2 (residues 37-108) before crossing back to complete Domain-1 (residues 109-224). During the early stages of the refinement, residual density was observed at the N-terminus equivalent to at least three additional residues. Inspection of the sequence of the expression vector suggested that three residues (G, H and S) from the linker were attached to the N-terminal methionine and these residues were added to the model. The apo2 structure has two independent V-Csn molecules in the asymmetric unit and the two molecules form a non-crystallographic dimer (Fig. 2e). Superposition of the two molecules of the apo2 dimer onto the apo1 structure gives a root-mean-squared deviation (RMSD) of 0.45 Å for both molecules. In the apo1 crystal form, the same dimer is observed albeit generated by the crystallographic symmetry of the C2 space group (Fig. 2e). This is consistent with the observation that V-Csn exists as a dimer in solution based on size exclusion chromatography, although there is no evidence that dimerization is required for enzyme activity. Formation of the dimer buries 1830 Å² (~9%) of the surface per monomer. The regions of contact involve the loop between

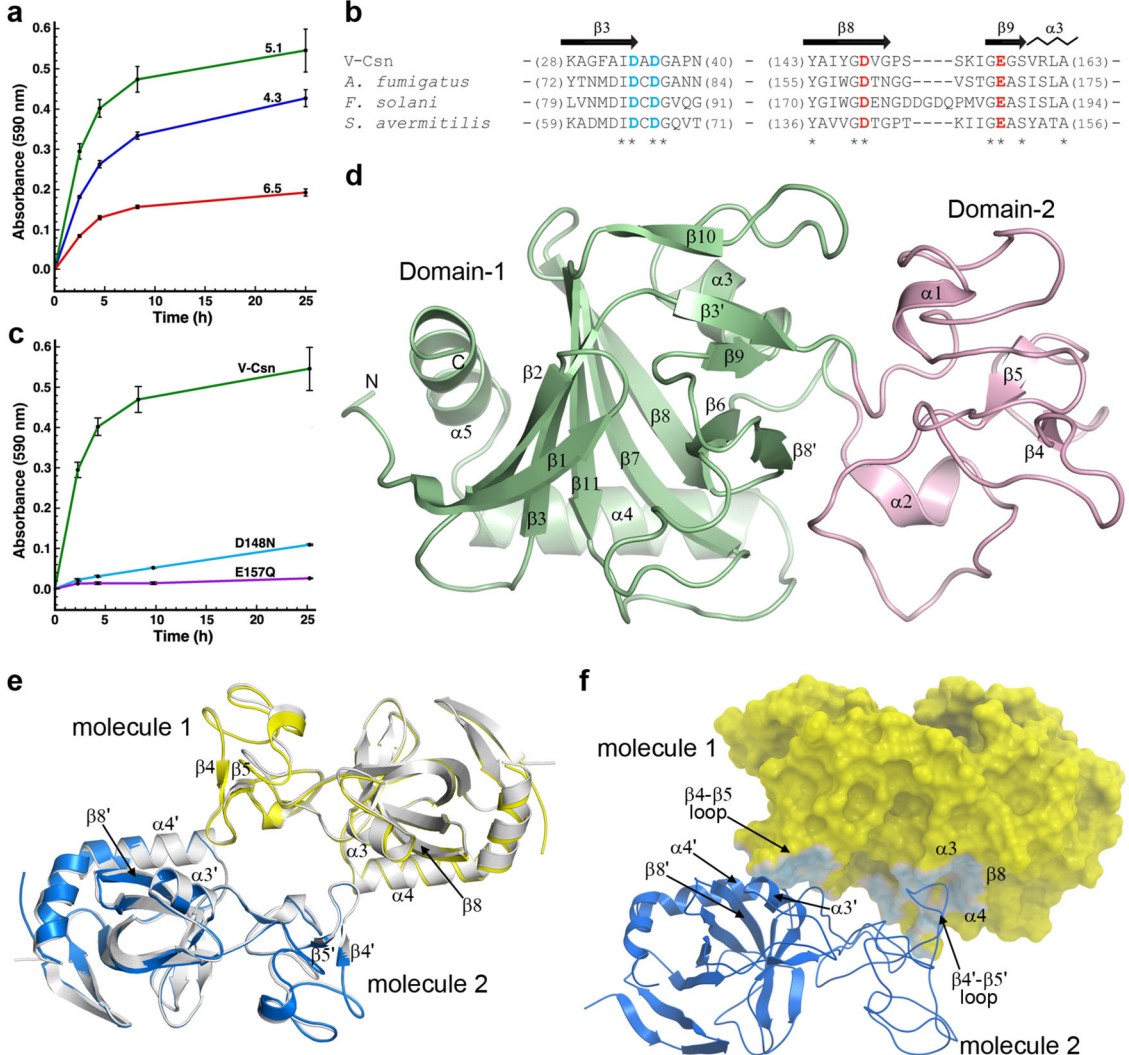

**Fig. 2 | The activity and structure of the soil viral chitosanase enzyme V-Csn. a** Activity of V-Csn was assayed by monitoring azurine absorbance at 590 nm following its release as soluble azurine-sugar fragments from an insoluble azurine cross-linked chitosan substrate (AZCL-chitosan). Reactions were performed at room temperature over a range of pH values in acetate buffer. The data represent the mean of replicate experiments ($n = 3$) with the standard deviation indicated by the error bars. Source data are provided as a Source Data file. **b** Partial sequence alignment of V-Csn with three chitosanase enzymes from *Aspergillus fumigatus*, *Fusarium solani* and *Streptomyces avermitilis*. The secondary structure for V-Csn is indicated above the alignment. Four conserved acidic residues potentially involved in catalysis are colored blue and red. Alignment of available sequences of GH75 family enzymes show that only six positions are universally conserved, these four acidic residues and two glycine residues, as indicated by the asterisks beneath the alignment. **c** Release of azurine by wild-type V-Csn and two constructs containing either a D148N or E157Q substitution, in acetate buffer at pH 5.1. The data represent the mean of replicate experiments ($n = 3$) with the standard deviation indicated by the error bars. Source data are provided as a Source Data file. **d** Ribbon representation of the apo1 form of the V-Csn enzyme with the two structural domains colored light green (Domain-1) and pink (Domain-2). The secondary structure nomenclature is given, along with the locations of the N- and C-termini. **e** Superposition of a dimer constructed from V-Csn apo1 (molecule 1, yellow ribbon) and a crystallographic symmetry partner (molecule 2, blue ribbon), and the V-Csn apo2 homodimer (gray ribbons). The secondary structure elements which comprise the dimer interface are indicated. Secondary structure elements labeled with a prime represent the symmetry related apo1 molecule. **f** The V-Csn apo2 dimer showing molecule 1 as a yellow molecular surface and molecule 2 as a blue ribbon. The contact area between the two molecules is shown in light blue on the yellow molecule 1 surface. Source data are available as a Source Data file.

β-strands β4 and β5 (in Domain-2) in one molecule slotting between helices α3 and α4 in the second molecule (Fig. 2f), linked via hydrogen bonding and hydrophobic interactions with residues from the two helices and strand β8.

Domain-1 is composed of a central six-stranded antiparallel twisted β-sheet made up of strands β1, β2, β3, β7, β8 and β10 (Fig. 3a, b). Two short strands (β6 and β9) pack against the concave face of the central β-sheet, and two helices (α4 and α5) wrap across the convex face of the sheet. A Dali search[25] using the isolated Domain-1 gives over 1000 hits with a Z-score greater than 5. An initial analysis of the top hits shows that Domain-1 has structural similarity with a diverse range of proteins that all have a common core domain comprising a double-*psi* β-barrel (DPBB) made up of

strands β3, β6, β7, β8, β9 and β10. These proteins include the plant defense proteins kiwellin[26,27], barwin[28] and carwin[29], the fungal phytotoxin cerato-platanin[30], the *Streptomyces* papain inhibitor (SPI)[31], domain 1 of the expansins (proteins which loosen plant cell walls)[32,33], the human ubiquitin regulatory domain of ASPL[34], and the carbohydrate hydrolyzing endoglucanases[35–37]. Barwin, carwin and the endoglucanases are classified as members of the glycosyl hydrolase GH45 family (https://pfam.xfam.org/family/PF02015), and the comparison with V-Csn shows that both the GH45 and GH75 enzymes bear a strong structural similarity. Both families, however, have no structural similarity with the GH46 enzymes, most of which are annotated as chitosanases but which have a two-domain α-helical architecture reminiscent of T4 lysozyme[38].

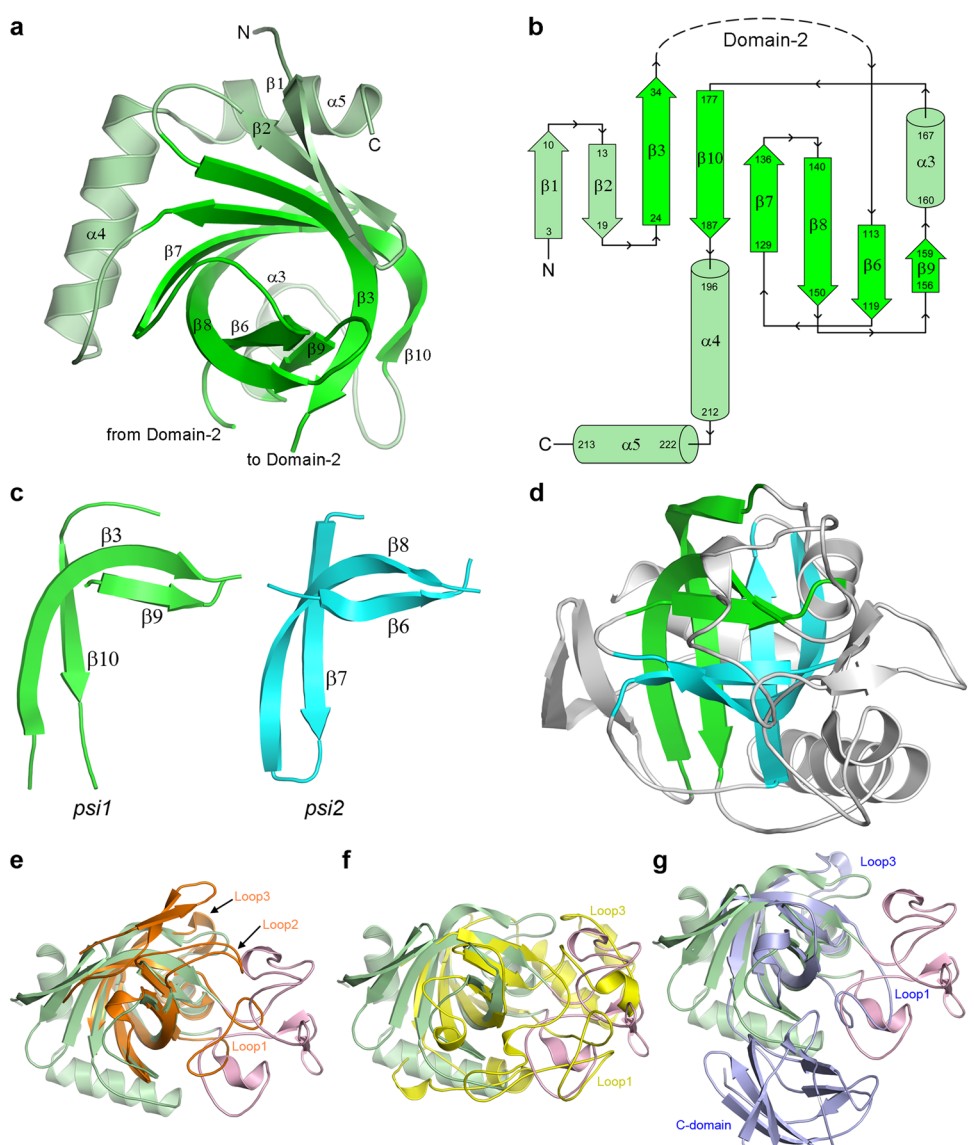

**Fig. 3 | Domain-1 structure and the DPBB motif. a** Domain-1 of the apo1 structure oriented to look down the double-*psi* β-barrel (DPBB) structural motif. The strands making up the DPBB are highlighted in bright green. **b** Topology diagram of Domain-1 highlighting the fold of the DPBB motif (bright green). **c** The two *psi* motifs (green and cyan) from the V-Csn apo1 structure, shown in approximately the same orientation so that their structural similarity is evident. **d** The two *psi* motifs in the context of the V-Csn apo1 structure. The green *psi* motif is in approximately the same orientation as in **c** and the cyan *psi* motif is rotated about 180° about an axis into the page. Together they make up the double-*psi* β-barrel (DPBB) structural motif. **e** Superposition of V-Csn Domain-1 (light green) and kiwellin from *Actinidia*

*chinensis* (PDB code 4PMK; orange ribbon). Loop1 and Loop2 in kiwellin coincide approximately with parts of Domain-2 (pink ribbon) of V-Csn. **f** Superposition of V-Csn Domain-1 (light green) and endoglucanase V from *Humicola insolens* (PDB code 4ENG; yellow ribbon). Loops 1 and 2 in endoglucanase V correspond approximately with Domain-2 (pink ribbon) of V-Csn. **g** Superposition of V-Csn Domain-1 (light green) and expansin from *Clavibacter michiganensis* (PDB code 4JCW; light purple ribbon). Loop1 of the expansin corresponds to part of V-Csn Domain-2 (pink ribbon), however, the C-terminal domain of the expansin (C-domain) bears no structural similarity to V-Csn.

The DPBB domain, comprising two interlocking *psi*-motifs, was first described for aspartate-α-decarboxylase, endoglucanase V, DMSO reductase and barwin[39]. In V-Csn each *psi*-motif is composed of two long antiparallel strands, one bent almost 90° such that the N-terminal half is almost orthogonal to the C-terminal half, and one single short strand running parallel with the C-terminal part of the long strand (Fig. 3c). The two *psi*-motifs (*psi1*; β3, β9 and β10, and *psi2*; β6, β7 and β8) are oriented relative to each other such that the two short strands (β6 and β9) form an antiparallel pair with a pseudo-twofold axis between them mapping *psi1* onto *psi2* (Fig. 3d). Superposition of several of the top DPBB-containing hits from Dali demonstrates the conserved topology of the two *psi*-motifs in these unrelated proteins (Fig. 3e–g). Several loop extensions decorate the DPBB domains of

these proteins. With respect to V-Csn strand numbering, these are: (i) Loop1, between strand β3 of the *psi1* motif and the strand β6 of the *psi2* motif, (ii) Loop2 between β8 of *psi2* and β9 of *psi1*, and (iii) Loop3 between strands β6 and β10 of the *psi1* motif. In V-Csn, the Loop1 extension encapsulates all of Domain-2, and in other proteins this loop varies in length and structure (Fig. 3e–g).

The V-Csn Domain-2 is very unusual in that it displays a distinct lack of secondary structure, and at first glance appears to be essentially unstructured (Fig. 2d). A Ramachandran plot analysis of the Domain-2 *phi/psi* angles (Fig. 4a) shows clustering of the main chain torsion angles into the favored α- and β- regions as would be expected for a well-folded protein. However, unlike a typical protein structure, Domain-2 seems to lack long continuous stretches of α-helical or β-

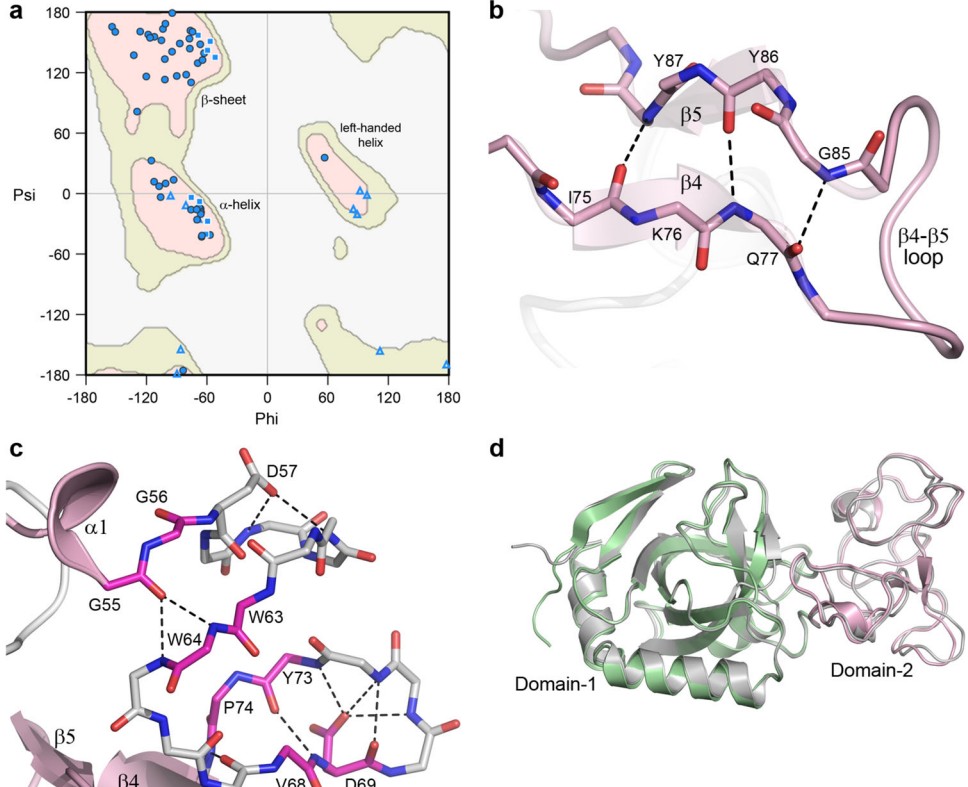

**Fig. 4 | Domain-2 structure. a** Ramachandran plot for Domain-2 of V-Csn apo1. The main chain torsion angles (*phi* and *psi*) are shown as light blue triangles (glycine), light blue squares (proline) and blue circles (all other residues). The three favored regions for β-sheets, α-helices and left-handed helices are colored pink. The additionally allowed regions are colored pale yellow. **b** Hydrogen bonding interactions between the two short β-strands, β4 and β5, in Domain-2. **c** Hydrogen bonding interactions in the 20-residue region in Domain-2 between helix α1 (a 3₁₀ helix) and strand β5 (shown as gray sticks for main chain atoms only). This region contains four residues designated as β-bridges, G56, W63, V68 and P74 (colored magenta). These residues are involved in main chain-main chain hydrogen bonds with other β-bridge residues and residues adjacent to them. The 20-residue region is folded into two hairpin loops, with an acidic residue (D57 and D69) in each loop stabilizing each hairpin through hydrogen bonding interactions with main chain amide nitrogen atoms. **d** Superposition of the predicted AlphaFold V-Csn structure (gray ribbon) and the V-Csn apo1 crystal structure (green and pink ribbons representing Domain-1 and Domain-2, respectively).

strand structure. Calculation of the secondary structure characteristics using multiple algorithms including DSSP (as implemented in PRO-CHECK and PyMOL) and the STRIDE server[40] all identify two single turn 3₁₀ helices (α1 and α2) and two short strands (β4 and β5), along with four individual residues annotated as β-bridges (residues G56, W63, V68 and P74). The two short β-strands run anti-parallel to each other and are connected via three hydrogen bonds (Fig. 4b). The four β-bridge residues are localized to a piece of polypeptide between helix α1 and strand β4 which is folded into two hairpin turns and held together by hydrogen bonding interactions between main chain atoms of these β-bridge residues, along with several side chain/main chain hydrogen bonds. (Fig. 4c). An AlphaFold prediction of V-Csn, made after the crystal structure had been completed, was remarkably close across the entire sequence (0.6 Å RMSD for 222 matching Cα atoms), including the uncharacterized Domain-2 (0.8 Å RMSD for 67 matching Cα atoms) (Fig. 4d).

### The active site
Inspection of the apo1 structure shows that the two residues identified as putative active site residues (D148 and E157) are located in a cleft between the two structural domains (Fig. 5a). Two additional acidic residues (D34 and D36) are also located in this cleft adjacent to D148, and these residues were also conserved in the other GH75 chitosanases (Fig. 2b). In V-Csn, the side chain of D148 makes hydrogen bonding interactions with the main chain amide nitrogen of A92 and the side chains of both D34 and D36 (Fig. 5b). Although the clustering of acidic residues like this is unusual, it is not unprecedented and occurs either

in metalloproteins where acidic side chains are brought close together by their roles in metal binding, or in enzyme active sites where they share protons[41]. At the pH of crystallization (4.6), it would be expected that most of these acidic residues would be protonated and thus essentially neutral, and calculation of the electrostatic surface within the active site cleft at this pH shows very little negative charge (Fig. 5c). At the pH optimum of the enzyme (5.1–5.5), however, the aspartate residues would be somewhat less protonated and there is significant negative charge within the cleft (Fig. 5d), which may be important for attracting the chitosan substrate into the pocket.

### Structures of two site-directed chitosanase mutants
Site-directed mutants D148 and E157 were constructed. In both cases the carboxylate was converted to the corresponding amide, generating the mutant proteins D148N and E157Q. The two mutant V-Csn proteins were crystallized under the same conditions as the wild-type protein, and their structures were determined by molecular replacement to high resolution. Superposition of the two mutant structures onto the wild-type apo1 structure gave RMSDs of 0.11 and 0.07 Å, respectively, for all Cα positions, suggesting very little conformational differences between the mutant and the wild-type structures. Co-crystallization of both mutant proteins with chitohexaose (a β-(1-4)-linked polymer of six D-glucosamine (GlcN) residues) gave a complex with the E157Q mutant only. The substrate was located in the inter-domain cleft (Fig. 5e), with three of the six GlcN residues (hereinafter named GlcN-1, GlcN-2 and GlcN-3) visible in $F_o$-$F_c$ electron density (Fig. 5f), oriented such that GlcN-1 is the reducing end. The GlcN-2 and

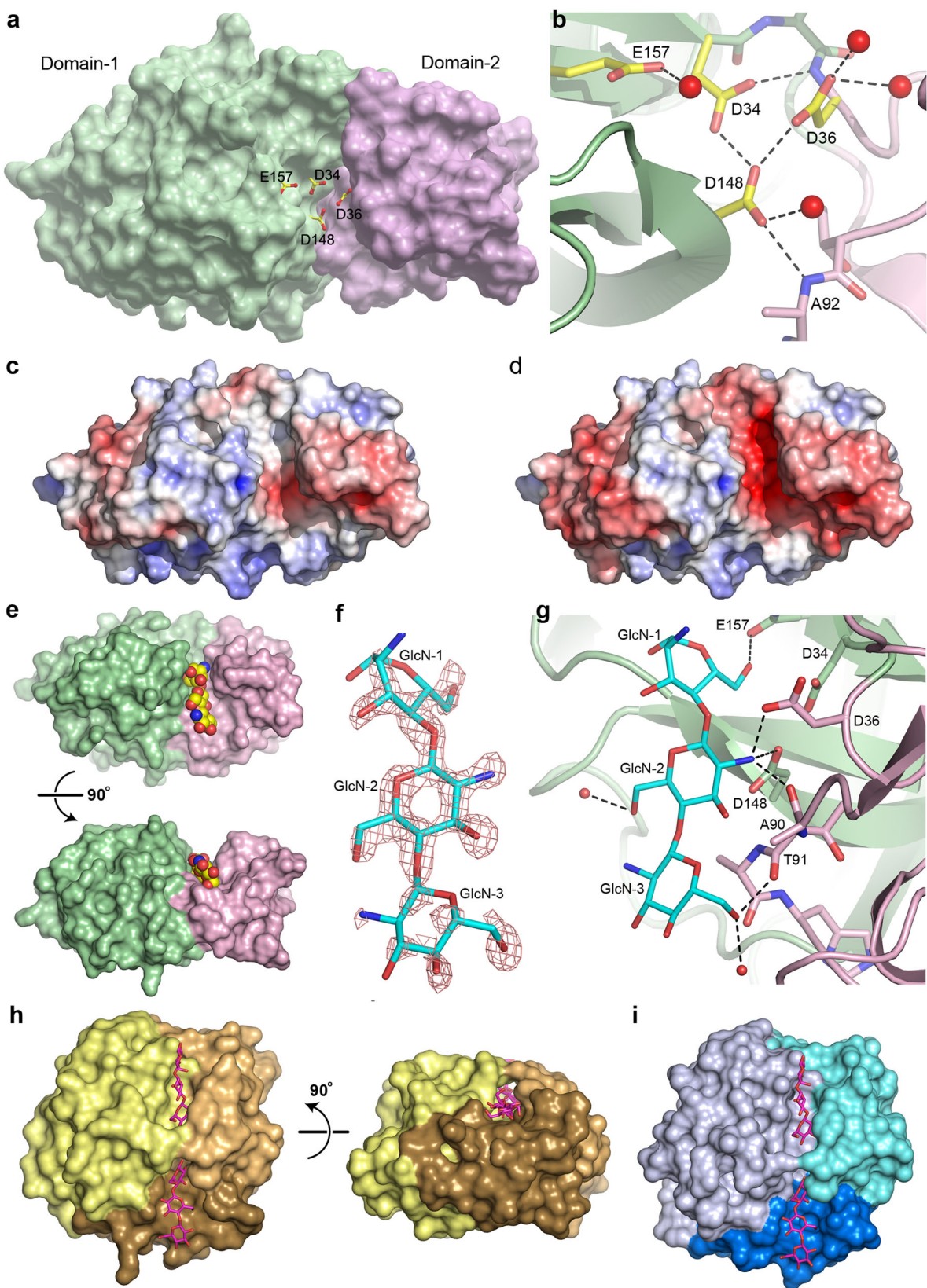

GlcN-3 residues are in a standard chair conformation, however, the density for the first observed residue (GlcN-1) suggested a distorted boat conformation.

The GlcN-1 residue is anchored by a single hydrogen bond between the O6 atom and the side chain of Q157 (Fig. 5f). The central GlcN residue (GlcN-2) makes hydrogen bonding interactions with the side chains of D148 and D36 via its free amine, along with a third to the backbone carbonyl oxygen of A90 from Domain-2. The GlcN-3 residue makes a hydrogen bonding interaction with the carbonyl oxygen of T91 via the O6 atom, and another to a water molecule. Although some additional $F_o$-$F_c$ density was observed at the non-reducing end of GlcN-3 (the O4 atom), the sparsity of the density did not allow for a fourth

**Fig. 5 | V-Csn active site and substrate complex. a** Solvent accessible surface representation of V-Csn apo1 structure with four conserved acidic residues (yellow and red sticks) clustering within the inter-domain cleft. Surface colored as for Fig. 2d: Domain-1, green; Domain-2, pink. **b** Close-up of putative active site highlighting the four conserved acidic residues. Hydrogen bonds shown as dashed black lines and water molecules as red spheres. **c** Electrostatic surface of V-Csn apo1 calculated at pH 4.6. Surface is contoured from −5 kT/e (red) to +5 kT/e (blue). **d** Electrostatic surface of V-Csn apo1 calculated at pH 5.5. Surface is contoured from −5 kT/e (red) to +5 kT/e (blue). **e** Solvent accessible surface representation of chitohexaose-V-Csn complex showing location of trisaccharide moiety (yellow, blue and red CPK spheres) in active site cleft. Orientation of molecule in top view is approximately the same as in **a**, and bottom view is rotated 90° to show side view. Enzyme colored by structural domains (Domain-1, green; Domain-2, pink). **f** Residual $F_o$-$F_c$ electron density (pink mesh) for bound substrate contoured at 2.5 σ. Electron density map was calculated following molecular replacement and prior to

the incorporation of substrate. Final refined trisaccharide molecule (GlcN-1, GlcN-2 and GlcN-3) is shown as cyan sticks. **g** Ribbon representation of chitohexaose-V-Csn complex with trisaccharide (cyan sticks) bound in the active site. Hydrogen bonds indicated by dashed black lines and water molecules as small red spheres (Domain-1, green; Domain-2, pink). **h** Solvent accessible surface representation of cellohexaose complex of *Humicola isolens* endoglucanase V (Cel45): DPBB domain, yellow; two loops for small sub-domains, brown (lower) and orange (upper). Cellohexaose molecule (magenta) is bound in narrow tunnel between DPBB domain and the two loops. View on right side of panel is rotated 90° to show the tunnel-like active site, in contrast to the open active site in V-Csn. **i** Solvent accessible surface representation of *Cryptopygus antarcticus* endoglucanase CaCel: DPBB domain, light blue; two loops are colored blue (lower) and cyan (upper). Location of substrate bonding cleft and tunnel indicated by cellohexaose molecule (magenta sticks) derived from the Cel45 complex in **h**.

GlcN to be modelled. The location of the trisaccharide fragment and the interactions it makes with the protein suggests that residues D36 and D148 form the −2 subsite[42] and play a key role in binding and orienting the substrate (in this case via the GlcN-2 residue). The E157 residue represents the −1 subsite and may serve as the nucleophilic group responsible for bond cleavage, assuming that hydrolysis occurred at the reducing end of GlcN-1.

Although the function of V-Csn Domain-2 is not fully understood, several pieces of evidence point to it being involved in the formation of the active site and in substrate binding. Comparison of the V-Csn structure with structures of GH45 family enzymes endoglucanase V (Cel45) from *Humicola insolens* (PDB code 3ENG)[35,36] and the endo-β −1,4-glucanase (CaCel45) from *Cryptopygus antarcticus* (PDB code 5H4U)[37] shows that the active site cleft in these glycosyl hydrolases is made up on one side by the DPBB domain and the loops which carry the catalytically important acidic residues, and on the other by Loop-1 (a long meandering loop in the GH45 enzymes and equivalent to the entire Domain-2 in V-Csn) and Loop-3 (a three-helix bundle in the GH45 enzymes) (Figs. 3e, 5h, i). Analysis of individual subsites in the cellohexose complex of Cel45 (PDB code 4ENG) shows that both the −1 and +2 subsites are formed partially by residues from Loop1 and Loop3 respectively, and these subsites which flank the +1/−1 cleavage site would be important for the correct orientation of the cellulose substrate prior to hydrolysis. In fact, in both Cel45 and CaCel45, the active site cleft walls form a tunnel through which the cellulose substrate is threaded (Fig. 5h, i).

As noted earlier, a loop in V-Csn Domain-2 (residues Ala90 and Thr91) is involved in formation of the −2 and −3 subsites (Fig. 5g) and would be expected to correctly orient this end of the chitosan substrate for optimal positioning of the −1/+1 cleavage site. Moreover, superposition of the cellohexoase-Cel45 complex onto the chitotriose-V-Csn complex and modelling of the cellohexose moiety into the V-Csn active site cleft (Fig. 6a, b) shows residues Asp50 and Asn54 from Domain-2 could form the +1 subsite, effectively anchoring this end of the substrate, with the +2 and +3 subsites most likely confined to Domain-1. Superposition of the AlphaFold predicted models for the nine viral chitosanases chosen for enzymatic function validation (Fig. 1) onto V-Csn shows that they all have an aspartate spatially equivalent to Asp50 in V-Csn, and four out of the nine have either an Asn or Asp in an equivalent location to Asn54. Although the cleft formed between Domain-1 and Domain-2 in V-Csn (Fig. 5e) is wider than the tunnel in the GH45 enzymes, the structural similarities between V-Csn and the GH45 enzymes point to Domain-2 playing a key role in the recognition, binding and optimal orientation of the chitosan substrate in V-Csn.

## Proposed mechanism

As noted earlier, biochemical and enzymological studies on some known GH75 chitosanases implicated two acidic residues (equivalent

to D148 and E157 in V-Csn) as being critically involved in catalysis[18,19,21]. It was established that the GH75 enzymes are endoglucanases[18] that invert the stereochemistry at the anomeric carbon, producing the α anomer of the oligosaccharide products[19,21], so it is likely that V-Csn is also an inverting enzyme. It is notable that in DPBB enzymes annotated as carbohydrate binding and/or hydrolyzing enzymes, the acidic residue at a position equivalent to E157 in V-Csn is universally conserved (either a glutamate or an aspartate), based upon the superposition of V-Csn with these DPBBs and the subsequent generation of a structure-based partial sequence alignment (Fig. 6c). Structural data from Cel45[35,36] and CaCel45[37] suggest that this acidic residue is the catalytic proton donor in the GH45 enzymes, and given the structural similarity of the DPBB domains in the GH45 and GH75 enzymes, it is highly likely that E157 is the catalytic proton donor in V-Csn. It should be noted that the GH45 enzymes are classified as endoglucanases which also lead to inversion of configuration at the anomeric carbon of the cleaved glycosidic bond[43].

The identity of the catalytic base is less clear, although based upon the same superposition and sequence alignment (Fig. 6c), Cel45 and CaCel have acidic residues near the N-terminus of the respective enzymes (D10 in Cel45 and D13 in CaCel) which have been identified as the general base accepting the proton during hydrolysis of the β(1,4) glycosidic bond[35,37]. These residues are structurally equivalent to D36 in V-Csn (Fig. 6d), which suggests that this residue may be acting as the catalytic base in the viral enzyme also. A schematic representation of the putative reaction mechanism for V-Csn is given in Fig. 6e. As previously noted, the fungal and bacterial GH75 enzymes have an aspartate residue at this same location (Fig. 2b). Conversely, other DPBB enzymes tentatively annotated as carbohydrate binding and/or hydrolytic enzymes lack an acidic residue equivalent to D34 or D36 (Fig. 6c) with the exception of the *Streptomyces* papain inhibitor protein (PDB code 5NTB)[31] and kiwellin (PDB code 4PMK)[26], yet they do have an aspartate structurally equivalent to D148 which may be acting as the catalytic base in these enzymes. Although the function of each of the four acidic residues in the V-Csn active site are not yet fully understood, their clustering within the cleft, and the corroborating evidence from the GH45 and GH75 enzymes suggest that it is highly likely that they will have roles in substrate binding (D34 and D148) and the catalytic mechanism (D36 and D157). Validation of the assignment of function must await further mutational and substrate binding studies.

## Ecological implications

To summarize, there are several ecological implications of this study. We identified putative AMGs from soil viral sequences by applying stringent screening criteria and inspections of up/down-stream coding regions, and conclusively demonstrate that at least some AMGs carried on soil viruses are functional. Our rigorous analyses not only resulted in a crystal structure of a soil viral AMG product, but also enabled us to

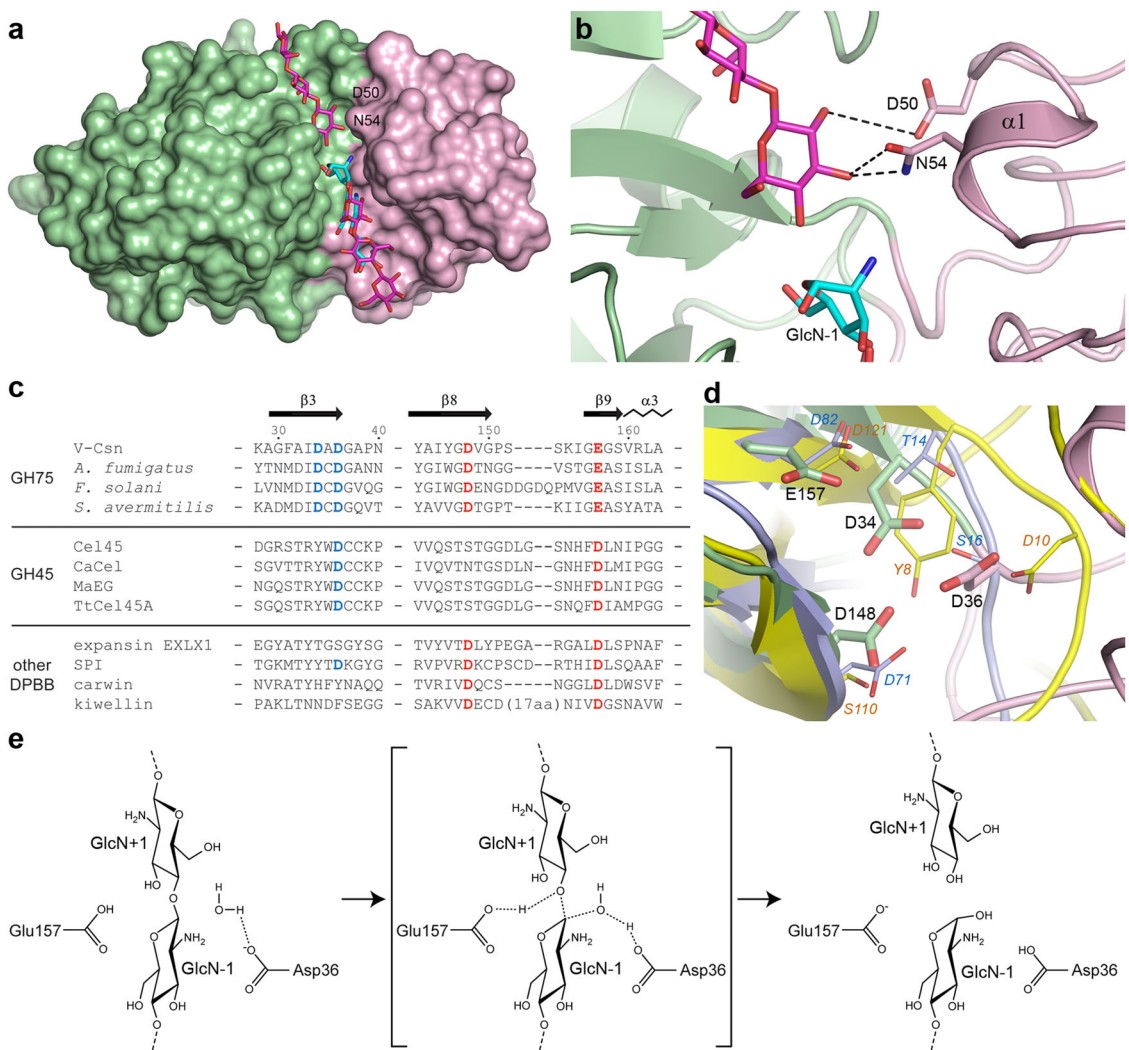

**Fig. 6 | Active site and reaction mechanism. a** Molecular surface representation of the E157Q-V-Csn mutant (green and pink domains) with cellohexaose (magenta sticks) modeled into the active site cleft based upon superposition of the chitohexaose-Cel45 complex (PDB code 4ENG). The three GlcN residues observed in the E157Q-VCsn complex are shown as cyan sticks. **b** The putative GlcN+1 recognition site in V-Csn based upon the superposition of the chitohexaose-Cel45 complex. **c** Partial structure-based sequence alignment of V-Csn with several enzymes containing DPBB domains. The alignment was determined from superpositions of the enzymes against V-Csn based upon their DPBB motifs. The four conserved acidic residues observed in the V-Csn active site are colored blue and red. Matching residues in the GH45 enzymes and other DPBB containing proteins are colored similarly when structurally equivalent. The sequences used are as follows: Cel45, endoglucanase V from *Humicola isolens* (Genbank P43316.1); CaCel,

endoglucanase from *Cryptopygus antarcticus* (Genbank ACV50415.1); MaEG, endoglucanase from *Melanocarpus albomyces* (Genbank CAD56665.1); TtCel45A, endoglucanase from *Thielavia terrestris* (Genbank 1182607135); EXLX1, expansin from *Bacillus subtilis* (Genbank O34918); SPI, Steptomyces papain inhibitor from *Streptomyces mobaraensis* (Genbank WP_004951535.1); carwin from *Carica papaya* (Genbank 4JP6_A); kiwellin from *Actinidia chinensis* (Genbank AGC39168.1). **d** Superposition of V-Csn (green and pink ribbons, green and pink sticks and black bold labels), Cel45 (yellow ribbon, thin yellow sticks and orange italic labels, PDB code 3ENG) and expansin EXLX1 (light blue ribbon, light blue sticks and cyan italic labels, PDB code 4FFT). The structures were superimposed by matching the DPBB motifs. **e** Schematic representation of the proposed glycoside bond cleavage reaction catalyzed by V-Csn. Glu157 is acting as the catalytic proton donor and Asp36 is acting as the catalytic base which accepts a proton from a water molecule.

propose the mode of action of this previously uncharacterized chitosanase enzyme in the GH75 family of glycosyl hydrolases. The chitosanase sequences that were included and compared revealed a phylogenetic distinction between viral chitosanases and those previously described in bacteria and fungi. However, because the viral chitosanases were subgroups within bacterial clades and the viruses detected with the chitosanase AMGs were bacteriophages, this suggests that they originated from bacteria. The soil viral chitosanases also formed subgroups that were distinct from their counterparts in aquatic systems. The V-Csn enzyme that we functionally and structurally characterized originated from a sequence from a forest soil (DOI 10.46936/10.25585/60000627, Supplementary Table 1). Forest soils are often characterized as having more fungi than other soil types[44]. This may be a reason for selection of viruses that carry the capacity to

help to decompose chitin - a major component of fungal cell walls and an important source of both carbon and nitrogen. The reason for selection of a virus that carries this capacity independently of its host is currently unknown. By analogy to marine systems where viruses carry AMGs that help to support energy generation via photosynthesis in their respective hosts[8,9,45], soil viruses may also help their hosts to decompose available carbon resources in soil as they become available.

## Methods

### Viral contig acquisition and chitosanase AMG detection
The Integrated Microbial Genomes and Virome (IMG/VR) database (v3.0)[46] was screened for sequences corresponding to predicted chitosanase genes. Viral contigs with genes annotated by a chitosanase

HMM (pfam07335) were first identified by applying a JGI viral detection pipeline[47]. For a more conservative functional assignment, the viral chitosanase sequences were further checked against annotation databases including EggNOG[48], the carbohydrate-active enzyme database (CAZY)[49] and the functional ontology assignments for metagenomes database (FOAM)[50] using hmmsearch (Hmmer v3.1b2)[51] and searching for sequence similarities to NCBI chitosanases using blastp (v2.9.0+)[52]. The putative viral chitosanases were then screened against a profile of lysozyme HMMs to remove the mis-annotated lysozymes (PF13702, PF00959, PF04965, PF18013, PF00062 and a self-curated lysozyme HMM[4] using the lysozyme sequences deposited at NCBI viruses (accessed on 16 November 2020).

For a confident assignment of the chitosanase genes as viral AMGs, the genomic content of the viral contigs carrying chitosanase genes screened from the above steps were inspected. Genes from viral contigs were predicted and translated using Prodigal (v2.6.3)[53]. The protein sequences were annotated by EggNOG bacterial and archaeal databases and three viral databases[4,54], in addition to the 7185 microbial-specific and 8773 viral-specific HMMs implemented in checkV (v0.7.0)[55]. The chitosanase AMG candidates were classified into five categories according to their gene positions on viral contigs and presence or absence of viral hallmark genes[4]: Category 0, viral hallmark genes (i.e., genes encoding viral structural proteins, terminases and integrases) both upstream and downstream; Category 1, viral-specific genes both upstream and downstream, plus viral hallmark genes either upstream or downstream; Category 2, viral-specific genes both upstream and downstream, but without viral hallmark genes; Category 3, viral-specific genes either upstream or downstream, but without viral hallmark genes; Category 4, located on an edge of the viral contig. Only viral contigs with high confidence scores (categories 0-2) for chitosanase AMGs were retained for subsequent analyses (Supplementary Table 1).

## Viral contig clustering and host prediction

The viral contigs with chitosanase AMGs were clustered with Viral RefSeq genomes (v201) based on a scored protein sharing matrix. A clustering network including pairwise interactions was generated by applying vConTACT using default parameters (v2.0.9.10)[56]. The soil viral contigs did not share sufficient genes with previously deposited reference viruses to enable a confident taxonomic assignment.

The putative hosts of the viral contigs that carried chitosanase AMGs were predicted using three published bioinformatic tools: 1) WIsH[57] (v1.0, best-hit), 2) VirHostMatcher[58] (v1.0.0, best-hit) and 3) Prokaryotic virus Host Predictor (PHP)[59] ('consensus'). The final host taxonomy of a viral contig was assigned when results from at least two of the three tools reached consensus.

## Phylogenetic analysis of chitosanases

To delineate the phylogenetic relatedness of the detected viral chitosanases to GH75 chitosanases in other taxa, a phylogenetic tree was constructed based on multiple sequence alignments of protein sequences of archaeal, bacterial, fungal and viral chitosanases. The tree was re-rooted using a bacteriophage lysozyme (YP_006987285.1). In order to cover the diverse genetic space across all domains of life, we first queried 'chitosanase' from NCBI protein database (https://www.ncbi.nlm.nih.gov/protein, accessed on Oct 11th, 2021) and further screened by the GH75 chitosanase pfam (PF07335). Sequences of the bacterial and fungal GH75 chitosanases used to identify key residues in the active sites were also included as part of refs. 18, 19. The reference sequences were then clustered at 70% amino acid identity to remove redundancy using CD-HIT (v4.8.1)[60] and the representative sequence of each cluster with length longer than 150 amino acids was included in the final reference set, resulting in two sequences from archaea, 230 from bacteria and 180 from fungi. The de-replicated viral chitosanases and the reference sequences were aligned using MAFFT with default

parameters (v7)[61]. The multiple sequence alignments (MSAs) were manually inspected and adjusted based on positions of the four key residues of the predicted active site across the viral and reference sequences. The region in the alignment is from the first conserved residue (predicted active site) to the last conserved residue. In addition, we retained adjacent residues that were well aligned where <10% of the sequences had a gap (Supplementary Data 1). The phylogenetic tree was built using RAxML (v1.0.1) with model specified LG+ G8+ F and 500 bootstraps[62].

## Protein expression and purification

The gene encoding for a putative soil viral chitosanase sequence (Ga0126380_1000012531: noted with a double asterisk in Fig. 1) was chemically synthesized (Twist Bioscience, San Francisco, CA) and inserted into the NdeI site of pET28a inclusive of a 20-residue extension at the N-terminus (MGSS**HHHHHH**SSG<u>LVPRGSH</u>) containing a poly-histidine metal affinity tag (bold) and thrombin protease cleavage site (underlined) in the primary amino acid sequence of the expressed protein. The recombinant plasmid was used to transform chemically competent *Escherichia coli* BL21(DE3) (Invitrogen, Carlsbad, CA) from which ~1 mL ~15% glycerol stocks (LB media, $OD_{600nm}$ = ~0.8) were prepared from a single colony and frozen (−80 °C) for future use. This glycerol stock was used to seed 25 mL of LB medium that was grown to an $OD_{600nm}$ of ~0.8 and then transferred to 750 mL of autoinduction LB medium[63] (2 L flasks, 200 rpm shaker, 0.34 ug/uL kanamycin, 37 °C). Upon reaching an $OD_{600nm}$ of approximately 1, the temperature was lowered to 30 °C. The cells were harvested ~16 h later (next day) by gentle centrifugation and then frozen (−80 °C). Cells were lysed by thawing the frozen pellet followed by sonication (~1 min) before and after three passes through a French Press (SLM Aminco, Rochester, NY). Following centrifugation, the protein in the soluble fraction was purified using a conventional two-step purification protocol: metal chelate affinity chromatography on a 20 mL Ni-Agarose 6 FastFlow column (GE Healthcare, Piscataway, NJ) followed by gel-filtration chromatography on a Superdex HiLoad 26/60 column (GE Healthcare, Piscataway, NJ)[64]. Fractions containing the target protein after the last column step were concentrated to 2–5 mg/mL (Protein Buffer: 100 mM NaCl, 20 mM Tris, 1 mM DTT, pH 7) and stored at 4 °C until used for crystallization or enzyme assays. Yields of 2– 4 mg purified protein were obtained per liter LB medium. The same protocol was applied to prepare two modified proteins each containing the point substitution D148N or E157Q. Mutagenesis was performed as follows[65]: Briefly, the first strand of the template plasmid containing the wild-type gene was nicked with Nt.BbvCI and digested with exonuclease I and III. Oligonucleotides (Thermo Fisher, Pleasanton, CA) were designed to introduce point mutations at desired locations and added in a ratio of 1:20 with the single stranded template DNA. The priming sequences were extended with Phusion polymerase. After resolving nicks with Taq DNA ligase, the second wild-type strand was nicked with Nb.BbvCI, digested with exonuclease I and regenerated by priming from a second oligonucleotide to create a complete mutagenized dsDNA molecule. All enzymes were obtained from NEB, Ipswich, MA. Assembled constructs were sequence verified (Pacific Biosciences Sequel IIe, PacBio, Menlo Park, CA).

## Chitosanase activity assays

Wildtype V-Csn and the two modified proteins were tested for *endo*-chitosanase activity using an azurine cross-linked (AZCL) chitosan substrate (AZCL-chitosan; Megazyme, Wicklow, Ireland)[66]. Stock solutions (1200 µg/mL) of each protein were prepared in Protein Buffer along with AZCL-chitosan suspensions (2500 µg/mL) at pH 4.3, 5.1, and 6.5 in 40 mM sodium acetate, 100 mM NaCl, 1 mM DTT. The reactions were performed in triplicate, at room temperature, by adding 17 µL of protein (20 µg) to 100 µL of AZCL-chitosan in a 500 µL Eppendorf tube. The tubes were agitated by rotation (40 rpm) in a

Multi-Purpose Tube Rotator (Fisher Scientific). Activity was monitored by pelleting the substrate with brief centrifugation and measuring the absorbance of released azurine-linked product at 590 nm (NanoDrop 2000c; Thermo Scientific) using a 2 µL aliquot. Blank reactions showed no release of azurine-linked product in the absence of protein and pH measurements before and after the reaction varied less than 0.1 pH unit.

## Crystallization, X-ray data collection and processing

Initial crystallization conditions for V-Csn were obtained using the hanging drop method employing the Top96 screen (Anatrace). Crystals were observed in multiple conditions. Crystals from several conditions were harvested and flash-cooled in liquid nitrogen in their respective crystallization conditions augmented with 20% ethylene glycol. The crystals were sent to SSRL for diffraction screening on beamline BL9-2. Three conditions gave crystals which diffracted to high resolution; condition #45 (0.2 M ammonium sulfate, 0.1 M sodium acetate pH 4.6, 30% MMePEG2000) in space group C2 with unit cell dimensions a = 108.84 Å, b = 47.63 Å, c = 45.55 Å, β = 97.8°, with one monomer in the asymmetric unit (AU); condition #38 (0.1 M citrate pH 5.5, 20% PEG3000) in space group C2 with unit cell dimensions a = 163.30 Å, b = 46.00 Å, c = 73.56 Å, β = 92.3°, with two monomers in the AU; and condition #20 (0.2 M ammonium sulfate, 0.1 M bis-tris pH 5.5, 25% PEG3350) in space group C2 with unit cell dimensions a = 80.47 Å, b = 35.76 Å, c = 80.66 Å, β = 118.5°, with one monomer in the AU.

Data sets were collected from single crystals in conditions #45 and #38. For the condition #45 crystal (designated apo1), 1800 0.2° images were collected on BL12-2 using X-rays at 17000 eV (0.72929 Å) and a Pilatus 6 M PAD detector running in shutterless mode. The images were processed with XDS (v. Feb 5 2021)[67] and scaled using AIMLESS[68]. The final data set comprised 174574 unique reflections to 0.89 Å resolution. For the condition #38 crystal (apo2), 1800 0.2° images were collected on BL9-2 using X-rays at 12658 eV (0.97946 Å) and a Pilatus 6 M PAD detector running in shutterless mode. The images were processed with XDS[67] and scaled using AIMLESS[68], and the final data set comprised 117982 unique reflections to 1.35 Å resolution. Additional data collection and processing statistics for both crystal forms are given in Table 1.

For experimental phasing, a KBr soaking solution was prepared by dissolving solid KBr in condition #45 crystallization buffer augmented with 25% glycerol until a saturated solution was obtained (as determined visually under a microscope). This solution was diluted with fresh buffer to form a 1/8 saturated crystal soaking solution. Several apo1 crystals were swished quickly in this solution and flash-cooled in liquid nitrogen. Diffraction data sets were collected from KBr-soaked apo1 crystals on beamline BL12-2 at the bromide edge (13,481 eV, 0.91967 Å). A total of 3600 images were collected with a rotation angle of 0.2°/image, using the inverse beam method and 20° wedges. The images were processed with XDS[67] and scaled using AIMLESS (v0.7.7)[68]. Additional statistics are given in Table 1. Initial analysis of the data indicated a strong anomalous signal from the bromide extending to approximately 1.7 Å resolution.

## Structure determination and refinement

The V-Csn structure was solved by Br-SAD (bromide single anomalous diffraction) methods implemented in PHENIX[22]. Following solvent flattening and density modification, the overall figure of merit (FOM) was 0.363 for 16 bromide sites. Autobuilding in PHENIX (v1.20.1-4487) generated a model comprising 221 out of 224 expected residues. Initial refinement with phenix.refine[24] gave an $R_{work}$ and $R_{free}$ of 0.158 and 0.187, respectively. The model was completed using COOT[23] and refined further with phenix.refine using the apo1 data to 0.89 Å resolution. Water molecules were added at structurally and chemically relevant positions, and the atomic displacement parameters for all

### Table 1 | Wild-type V-Csn data collection statistics*

|  | V-Csn apo1 | V-Csn apo1 Br-SAD | V-Csn apo2 |
|---|---|---|---|
| Space group | C2 | C2 | C2 |
| Resolution range (Å) | 37.2-0.89 (0.91-0.89) | 37.2-1.60 (1.63-1.60) | 38.1-1.35 (1.37-1.35) |
| Unit cell |  |  |  |
| - a, b, c (Å) | 108.84, 47.63. 45.55 | 108.66, 47.53. 45.62 | 163.30, 46.00. 73.56 |
| - β (°) | 97.8 | 97.9 | 92.3 |
| Mathews coefficient | 2.45 (49.7% solvent) | 2.44 (49.6% solvent) | 2.89 (57.4% solvent) |
| Molecules in the A.U. | 1 | 1 | 2 |
| Reflections (observed/unique) | 1148940/174574 | 400825/30343 | 803185/117982 |
| $R_{meas}$ # (%) | 0.093 (1.265) | 0.107 (0.597) | 0.060 (1.166) |
| $R_{pim}$ # (%) | 0.036 (0.549) | 0.029 (0.192) | 0.023 (0.454) |
| I/σ | 11.4 (1.6) | 19.8 (4.3) | 16.1 (1.5) |
| Completeness (%) | 99.1 (91.3) | 99.2 (90.6) | 98.4 (95.4) |
| CC½ $ | 0.998 (0.609) | 0.999 (0.896) | 1.0 (0.847) |
| Multiplicity | 6.6 (5.0) | 13.2 (9.3) | 6.8 (6.3) |
| Wilson B (Å²) | 5.3 | 12.4 | 15.4 |
| Anomalous completeness | - | 99.0 (89.5) | - |
| Anomalous multiplicity | - | 6.7 (4.7) | - |
| $CC_{anom}$ & | - | 0.415 | - |
| MSAN ‡ | - | 1.40 | - |

*Numbers in parentheses relate to the highest resolution shell. #$R_{meas}$ is the redundancy-independent merging R factor and $R_{pim}$ is the precision-indicating merging R factor[71]. $Percentage of correlation between intensities from random half-sets of data[72]. &Correlation of $\Delta I_{anom}$ from two random half-sets[73]. ‡MSAN is the mid-slope of the anomalous normal probability plot. Values >1 indicate significant anomalous signal[73].

### Table 2 | Wild-type V-Csn structure refinement statistics

|  | V-Csn apo1 | V-Csn apo2 |
|---|---|---|
| PDB Code | 7TVL | 7TVM |
| Resolution range (Å) | 37.2–0.89 | 38.1–1.35 |
| Reflections used, total/free | 174427/8555 | 117783/5888 |
| Working R-factor/$R_{free}$* | 0.1198/0.1307 | 0.1429/0.1742 |
| Total atoms |  |  |
| - protein | 1811 | 3442 |
| - solvent | 398 | 566 |
| B factors |  |  |
| - protein (Å²) | 7.30 | 19.9 |
| - solvent (Å²) | 22.3 | 34.0 |
| rms deviation |  |  |
| - bonds (Å) | 0.005 | 0.005 |
| - angles (°) | 0.937 | 0.815 |
| Ramachandran plot# |  |  |
| - residues in preferred regions (%) | 99.1 | 99.3 |
| - outliers | 0 | 0 |
| Molprobity score# | 1.01 (96th percentile) | 0.77 (100th percentile) |

*$R_{free}$ was calculated with 5% of the reflections. #Calculated by MOLPROBITY[74].

atoms in the structure were refined isotropically. The apo2 structure was solved by molecular replacement using the program MOLREP (v11.9.02)[69] from the CCP4 suite[70], using the refined apo1 structure as the search model. Final refinement statistics for the two apo-V-Csn structures are given in Table 2.

## Chitosanase mutant and substrate structures

V-Csn mutants D148N and E157Q were screened for crystallization using conditions #20, #38 and #45, and crystals were observed in all three. Diffraction data sets were collected from single D148N and E157Q crystals from condition #45. For the D148N crystals, 1800

## Table 3 | V-Csn mutant data collection and refinement statistics*

|  | V-Csn-D148N | V-Csn-E157Q | V-Csn-E157Q-chitohexaose |
|---|---|---|---|
| *Data Collection* | | | |
| Resolution range (Å) | 37.2-1.20 (1.22-1.20) | 37.1-1.15 (1.17-1.15) | 37.1-1.30 (1.32-1.30) |
| Unit cell | | | |
| - a, b, c (Å) | 108.76, 47.45. 45.43 | 108.58, 47.43. 45.46 | 108.94, 47.42, 45.40 |
| - β (°) | 98.0 | 97.9 | 97.8 |
| Reflections (observed/unique) | 471537/70261 | 546974/76188 | 372782/55448 |
| $R_{meas}$ (%)# | 0.071 (0.702) | 0.050 (0.361) | 0.090 (0.964) |
| $R_{pim}$ (%)# | 0.027 (0.285) | 0.019 (0.135) | 0.034 (0.380) |
| I/σ | 16.1 (2.6) | 22.1 (5.1) | 11.7 (1.8) |
| Completeness (%) | 98.1 (88.4) | 94.0 (88.5) | 97.8 (87.9) |
| CC½ $ | 0.999 (0.970) | 1.0 (0.964) | 0.999 (0.837) |
| Multiplicity | 6.7 (5.7) | 7.2 (7.0) | 6.7 (6.0) |
| Wilson B (Å²) | 8.9 | 7.1 | 11.3 |
| *Refinement* | | | |
| PDB Code | 7TVN | 7TVO | 7TVP |
| Reflections used, total/free | 70244/3400 | 76177/3757 | 55326/2759 |
| Working R-factor/$R_{free}$ & | 0.1251/0.1489 | 0.1185/0.1350 | 0.1319/0.1640 |
| Total atoms | | | |
| - protein | 1742 | 1762 | 1731 |
| - solvent | 310 | 300 | 330 |
| B factors | | | |
| - protein (Å²) | 10.6 | 9.24 | 12.5 |
| - solvent (Å²) | 28.7 | 26.2 | 28.2 |
| *rms deviations* | | | |
| - bonds (Å) | 0.005 | 0.005 | 0.005 |
| - angles (°) | 0.892 | 0.914 | 0.864 |
| Ramachandran plot‡ | | | |
| - preferred regions (%) | 98.2 | 98.2 | 98.2 |
| - outliers | 0 | 0 | 0 |
| Molprobity score‡ | 0.97 (99th percentile) | 0.61 (100th percentile) | 1.05 (99th percentile) |

*Numbers in parentheses relate to the highest resolution shell. In all cases the crystals are isomorphous with V-Csn Apo1 crystals. #$R_{meas}$ is the redundancy-independent merging R factor and $R_{pim}$ is the precision-indicating merging R factor[71]. $Percentage of correlation between intensities from random half-sets of data[73]. &$R_{free}$ was calculated with 5% of the reflections. ‡Calculated by MOLPROBITY[74].

images (0.2° rotation/image) were collected on BL12-2, and the data processed and scaled with XDS[67] and AIMLESS[68]. For the E157Q crystal, 1850 images were collected on BL12-2, and the data processed and scaled with XDS[67] and AIMLESS[68]. Data collection statistics are given in Table 3. Both structures were solved by molecular replacement with MOLREP[69] using the refined wild-type V-Csn structure as the starting model, with all water molecules removed. The D148N and E157Q structures were refined with phenix.refine[24], and final statistics are also given in Table 3.

The E157Q-substrate complex was prepared by dissolving 0.06 mg of chitohexaose (Biosynth) in 10 uL of E157Q at 3.3 mg/ml, giving a final chitohexaose concentration of around 5 mM. The complex was incubated at 4 °C for 1 h prior to setting up sitting drops against crystallization condition #45. The crystallization drops were streak-seeded several hours after setup and crystals of the complex were observed in all drops overnight. The crystals were morphologically similar to wild-type and mutant crystals grown under the same

conditions. The crystals were transferred into crystallization buffer augmented with 25% glycerol, and flash-cooled in liquid nitrogen. Diffraction data were collected at BL12-2. A total of 1800 images were collected, and the data processed and scaled with XDS[67] and AIMLESS[68]. The E157Q-substrate complex structure was solved by molecular replacement with MOLREP[69] using the refined wild-type V-Csn structure with all water molecules removed as the starting model, and refined with phenix.refine[24]. Data collection and refinement statistics are given in Table 3.

### Structure modeling by AlphaFold2
The AlphaFold structure predictions were run using either a locally-installed version of the software retrieved from the official GitHub repository (https://github.com/deepmind/alphafold) or the Google collaborative AlphaFold notebook (https://colab.research.google.com/github/sokrypton/ColabFold/blob/main/AlphaFold2.ipynb). Solvent accessible surfaces were calculated with PyMOL (v2.5.2) (Schrodinger) and ICM-Pro (v3.8-6a) (Molsoft), using a probe radius of 1.4 Å (equivalent to the radius of a single water molecule). The electrostatic surfaces were generated with the Adaptive Poisson-Boltzmann Solver (APBS) plugin for PyMOL (v2.5.2).

### Reporting summary
Further information on research design is available in the Nature Research Reporting Summary linked to this article.

## Data availability
The data that support this study are available from the corresponding authors upon reasonable request. The viral sequence data used for Fig. 1 are publicly available on the JGI website [https://img.jgi.doe.gov/cgi-bin/vr/main.cgi] with no use restriction according to the JGI data policy. The atomic coordinates and structure factors for the protein structures have been submitted to the RSCB Protein Data Bank (PDB) under accession codes 7TVL (V-Csn apo1), 7TVM (V-Csn apo2), 7TVN (V-Csn-D148N), 7TVO (V-Csn-E157Q), and 7TVP (V-Csn-E157Q chitohexaose complex). The source data underlying Fig. 2a, c are provided as Source Data files. Source data are provided with this paper.

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

## Acknowledgements

This work was supported by the Department of Energy (DOE) Office of Biological and Environmental Research (BER) and is a contribution of the Scientific Focus Area "Phenotypic response of the soil microbiome to environmental perturbations" to JKJ and KSH. A portion of this work was performed on a project award (https://doi.org/10.46936/cpcy.proj.2021.60161/60000437) under the FICUS program to JEM and used resources at the DOE Joint Genome Institute (JGI) and the Environmental Molecular Sciences Laboratory (EMSL), which are DOE Office of Science User Facilities. Both facilities are sponsored by the Biological and Environmental Research program and operated under Contract Nos. DE-AC02-05CH11231 (JGI) and DE-AC05-76RL01830 (EMSL). The crystal structures were determined at the Stanford Synchrotron Radiation Lightsource (SSRL). SSRL is a National User Facility operated by Stanford University on behalf of the U.S. Department of Energy, Office of Basic Energy Sciences under Contract No. DE-AC02-76SF00515. The SSRL Structural Molecular Biology Program is supported by the Department of Energy, Office of Biological and Environmental Research, and the National Institutes of Health (NIGMS) by Grant Number P30GM133894. Preliminary results of enzymatic function were provided by Gregor Tegl and Stephen Withers from the University of British Columbia.

## Author contributions

R.W. carried out the majority of the bioinformatics analyses. C.A.S. determined the crystal structures of the chitosanase enzymes and elucidated their mechanisms. GWB expressed the chitosanase proteins, screened crystallization conditions and performed activity assays. I.K.B. and Y.Y. carried out the synthesis and cloning of chitosanase genes and mutants. N.C.K. provided the database retrieval and DOIs of publicly available chitosanase sequences. D.P.-E., J.R.C. and C.A.S. performed the AlphaFold predictions of protein structure. K.S.H., J.K.J., J.E.M. and C.A.S. funded the research. J.K.J. coordinated the study. All authors contributed to writing of the manuscript.

## Competing interests

The authors declare no competing interests.
