## [Peer Review File · Nature Communications]

Structural characterization of a soil viral auxiliary metabolic gene product – a functional chitosanaseReviewers' Comments:

Reviewer #1:

Remarks to the Author:

Wu et al. show for the first time that a chitosanase of viral origin is a potentially active protein and resolve its structure. This is a much-needed advance in the field of viral ecology in general, as very few viral auxiliary metabolic genes have been demonstrated to actually have the potential to be active during infection.

This manuscript is very well written and clear. The figures are well executed and I only have a minor comment on them.

I did not find any overstating of results in the discussion.

All in all, this paper is ready for publication with minor revisions in my opinion.

However, the protein characterization results lie beyond my area of expertise, so I will not comment on them other than they are well written.

While the data from which the active chitosanase was predicted is public, I think it would be wise to acknowledge the original owners of the data, especially given the series of problematic uses of JGI data which has not yet been used in publications by the actual authors of the proposals. I'm not saying this is the case here, but I prefer to err on the side of caution when using mined data, especially since the proposal the data came from is about soil viruses. This the data was used in a recent publication: <https://doi.org/10.1016/j.soilbio.2022.108569>. Please cite it, even if only in the supplemental material.

Unless there is a citation limit, I would probably cite the papers that showed that psbA is active during infection: <https://doi.org/10.1038/nature04111>, <https://doi.org/10.1111/j.1462-2920.2005.00969.x>

Line 69: I think this statement could use a citation

Fig. 1: "tree tips" are called leaves. Could you please add a space to the node colors legend title? i.e., Phylogenetic group

I am finding it hard to differentiate between archaeal and viral (aquatic) colors. Are there any archaeal proteins in there?

Line 98: Maybe it's just me, but when I read the phrase "deep clades" I immediately think about deep branch attraction and tend to dismiss it. You have very well-formed clades that aren't deep, which supports them being real rather than an artifact. Hence, I just wouldn't use the word deep here.

Line 106-107: Given the high mutation rates of viruses, I'd be curious to know the ratio of synonymous to non-synonymous mutations in the active site (or even in general in these viral chitosanases). If there are a lot more synonymous mutations that would be another way to support the claim that they could be active during infection.

Another way to strengthen the ecological implications of the results is to add a line or two discussing up/down-stream genes from the chitosanase in the viral genomes. One of the main concerns I had when I started reading this paper was whether this is an actual viral genome, or just a misassembly. I appreciated the coding of genes in sup. table 1, and since you've already done this analysis – I would just mention it as I think it supports your results. I see this is mentioned in the methods. I would add it to the main text.

Methods:

Are any of the phages identified as prophages? Or are they all putatively lytic?

Were the multiple alignments trimmed? Usually, they are with a tool that removes positions represented in less than a certain cutoff of proteins, and removes overhangs. Building a tree on untrimmed sequences may be unreliable.

I also don't think that many people trust FastTree as much as other tree building tools. I would suggest trying IQtree or RaxML to verify the clades.

Reviewer #2:

Remarks to the Author:

Mining the auxiliary metabolic genes (AMGs) source from the metagenomes has been quite hot recently. The authors picked up 10 hits from the chitosanase-like AMGs, but most were expressed in the insoluble form, and only one was characterized. Further, the V-Csn was analyzed by activity assay and crystallography. The manuscript is well written and easy to follow.

Although they declared that less soil viral AMG had been expressed and characterized, the function of soil viral AMG is easy to expect if all the essential residues are conserved in the active site after the sequence alignment. As for the crystal structure, the authors focused on the characters of the structure itself but lacked the structure-functional analysis. For example, what is the function of the Novel Domain (Supplementary Information Fig. 2a)? After obtaining the complex structure and pointing out the key residues, readers may be curious about how those residues affect the activity. All of the above indicates that the manuscript does not provide much information and might not be suitable for Nat. Comm. Since the central part of the manuscript is the description of the crystal structure, it may be better for a specialized journal such as Structure, Journal of Molecular Biology or Journal of Structure Biology.

Other comments:

In Page 9, adding a figure here will be better to describe the proposed mechanism.

Fig. 1 brought limited information for readers as the structures or models are too small to compare.

Page 5 line 153, Obtaining the kinetic parameters will be more conclusive compared to the single-point measurement.

Author's Rebuttal

Reviewer #1 (Remarks to the Author):

Wu et al. show for the first time that a chitosanase of viral origin is a potentially active protein and resolve its structure. This is a much-needed advance in the field of viral ecology in general, as very few viral auxiliary metabolic genes have been demonstrated to actually have the potential to be active during infection.

This manuscript is very well written and clear. The figures are well executed and I only have a minor comment on them.

I did not find any overstating of results in the discussion.

All in all, this paper is ready for publication with minor revisions in my opinion.

However, the protein characterization results lie beyond my area of expertise, so I will not comment on them other than they are well written.

Authors' response: We thank the reviewer for these positive comments.

While the data from which the active chitosanase was predicted is public, I think it would be wise to acknowledge the original owners of the data, especially given the series of problematic uses of JGI data which has not yet been used in publications by the actual authors of the proposals. I'm not saying this is the case here, but I prefer to err on the side of caution when using mined data, especially since the proposal the data came from is about soil viruses. This the data was used in a recent publication: <https://doi.org/10.1016/j.soilbio.2022.108569>. Please cite it, even if only in the supplemental material.

Unless there is a citation limit, I would probably cite the papers that showed that psbA is active during infection: <https://doi.org/10.1038/nature04111>, <https://doi.org/10.1111/j.1462-2920.2005.00969.x>

Authors' response: As mentioned by the reviewer, we were careful to only use public data that had no use restrictions according to the JGI data use policy. We also checked the recent paper (<https://doi.org/10.1016/j.soilbio.2022.108569>) and confirmed that we did not use this data. We thank the reviewer for pointing us to the other references. We added these in the revision.

Line 69: I this this statement could use a citation

Authors' response: A relevant citation was added.

Fig. 1: "tree tips" are called leaves. Could you please add a space to the node colors legend title? i.e., Phylogenetic group

I am finding it hard to differentiate between archaeal and viral (aquatic) colors. Are there any archaeal proteins in there?

Authors' response: We revised the figure according to these suggestions.

Line 98: Maybe it's just me, but when I read the phrase "deep clades" I immediately think about deep branch attraction and tend to dismiss it. You have very well-formed clades that aren't deep, which supports them being real rather than an artifact. Hence, I just wouldn't use the word deep here.

Authors' response: We agree and removed the word 'deep'.

Line 106-107: Given the high mutation rates of viruses, I'd be curious to know the ratio of synonymous to non-synonymous mutations in the active site (or even in general in these viral chitosanases). If there are a lot more synonymous mutations that would be another way to support the claim that they could be active during infection.

Authors' response: We agree that this is interesting and we performed an additional analysis. As suggested by the reviewer, we calculated the relative frequency of each residue at the four putative active sites predicted for the viral chitosanases. The figure and related text were updated accordingly.

Another way to strengthen the ecological implications of the results is to add a line or two discussing up/down-stream genes from the chitosanase in the viral genomes. One of the main concerns I had when I started reading this paper was whether this is an actual viral genome, or just a misassembly. I appreciated the coding of genes in sup. table 1, and since you've already done this analysis – I would just mention it as I think it supports your results. I see this is mentioned in the methods. I would add it to the main text.

Authors' response: We added this to the main text as suggested.

Methods:

Are any of the phages identified as prophages? Or are they all putatively lytic?

Were the multiple alignments trimmed? Usually, they are with a tool that removes positions represented in less than a certain cutoff of proteins, and removes overhangs.

Building a tree on untrimmed sequences may be unreliable.

Authors' response: The tree was built based on multiple sequence alignments of the trimmed chitosanases that were called from viral contigs. We specified details in the methods section accordingly. The sequence alignment was manually inspected to better align the predicted active sites.

I also don't think that many people trust FastTree as much as other tree building tools. I would suggest trying IQtree or RaxML to verify the clades.

Authors' response: As suggested by the reviewer, we used RAxML-NG (v. 1.0.1) with 500 bootstraps as suggested. The Methods Section was updated accordingly.

Reviewer #2 (Remarks to the Author):

Mining the auxiliary metabolic genes (AMGs) source from the metagenomes has been quite hot recently. The authors picked up 10 hits from the chitosanase-like AMGs, but most were expressed in the insoluble form, and only one was characterized. Further, the V-Csn was analyzed by activity assay and crystallography. The manuscript is well written and easy to follow.

Authors' response: We thank the reviewer for these positive comments.

Although they declared that less soil viral AMG had been expressed and characterized, the function of soil viral AMG is easy to expect if all the essential residues are conserved in the active site after the sequence alignment. As for the crystal structure, the authors focused on the characters of the structure itself but lacked the structure-functional analysis. For example, what is the function of the Novel Domain (Supplementary Information Fig. 2a)? After obtaining the complex structure and pointing out the key residues, readers may be curious about how those residues affect the activity.

Authors' response: As for the novel domain (Domain-2) some discussion as to its probable function has been added, along with new figures in the Extended Data (Extended data Fig. 5e and 5f, and Extended Data Fig. 6a and 6b)

All of the above indicates that the manuscript does not provide much information and might not be suitable for Nat. Comm. Since the central part of the manuscript is the description of the crystal structure, it may be better for a specialized journal such as Structure, Journal of Molecular Biology or Journal of Structure Biology.

Other comments:

In Page 9, adding a figure here will be better to describe the proposed mechanism.

Authors' response: A figure has been added to the Extended data: FigS6e

Fig. 1 brought limited information for readers as the structures or models are too small to compare.

Authors' response: We reorganized Fig1 to increase the size of the models.

Page 5 line 153, Obtaining the kinetic parameters will be more conclusive compared to the single-point measurement.

Authors' response: The assay described in the paper using a commercially available insoluble chitosan with an attached chromophore was selected as a simple means of determining whether chitosanase activity (measured as absorbance in the visible range as soluble chromophore-linked chitosan oligosaccharides are released) was present, but it is not suitable for determination of kinetic parameters.

Reviewers' Comments:

Reviewer #1:

Remarks to the Author:

For the most part, the authors addressed my comments to my satisfaction.

The one exception is that trimming of the MAFFT multiple alignment is still missing (i.e. throwing out positions where >50% of sequences have a gap). This is a crucial step that determines the quality of the presented protein tree. A good chunk of the paper is based on that tree, so I want to verify that this point is made in the paper.

Aside from that, I only have a couple of very minor suggestions.

1. I gather from the tree that the active V-Csn had the D-D-D-E residues. I wish that was spelled out early on when discussing it (lines 84-85 or 154-156).

2. I don't see any mention in the main text of the host taxonomy or viral clustering described in the methods. If you can use the host taxonomy to say something about V-Csn - say it. If you can't, you could say that you were unable to identify a potential host. Otherwise, I think this section shouldn't be in the paper.

3. I'd understand if you're done adding stuff, but since you mention marine chitosanase, this preprint just came out that is very relevant. Maybe you could mention it around lines 70-71:
<https://www.biorxiv.org/content/10.1101/2022.06.23.497379v1>

Reviewer #2:

Remarks to the Author:

The authors explained all of my concerns related to the manuscript. I agree the publication at Nat. Comm.

Author's Rebuttal

Reviewer #1 (Remarks to the Author):

For the most part, the authors addressed my comments to my satisfaction.

The one exception is that trimming of the MAFFT multiple alignment is still missing (i.e. throwing out positions where >50% of sequences have a gap). This is a crucial step that determines the quality of the presented protein tree. A good chunk of the paper is based on that tree, so I want to verify that this point is made in the paper.

Authors' response: We clarified this point in the methods section as follows: "The region in the alignment is from the first conserved residue (predicted active site) to the last conserved residue. In addition, we retained adjacent residues that were well aligned where <10% of the sequences had a gap."

Aside from that, I only have a couple of very minor suggestions.

1. I gather from the tree that the active V-Csn had the D-D-D-E residues. I wish that was spelled out early on when discussing it (lines 84-85 or 154-156).

Authors' response: We added "with the predicted D-D-D-E active site residues." to the sentence on line 144 that refers to V-Csn.

2. I don't see any mention in the main text of the host taxonomy or viral clustering described in the methods. If you can use the host taxonomy to say something about V-Csn - say it. If you can't, you could say that you were unable to identify a potential host. Otherwise, I think this section shouldn't be in the paper.

Authors' response: We thank the reviewer for noticing this omission. We performed host prediction using multiple host prediction tools such as WIsH, VirHostMatcher and PHP (details in Methods) and assigned proteobacteria as the potential host. We now added a sentence about the host taxonomy as suggested on line 147 and refer to Supplementary Information Table 1 as follows: "The virus that carried V-Csn was predicted to be a Proteobacteria phage (Supplementary Information Table 1)."

3. I'd understand if you're done adding stuff, but since you mention marine chitosanase, this preprint just came out that is very relevant. Maybe you could mention it around lines 70-

71: <https://www.biorxiv.org/content/10.1101/2022.06.23.497379v1>

Authors' response: We really enjoyed reading this preprint. We added a reference to this manuscript at two places in the introduction as follows: "Picocyanobacteria utilize chitin that is mainly produced by arthropods as a key nutrient source in open oceans (ref 14). Chitin can also accumulate in soils because it is a component of fungal cell walls and insect exoskeletons. Following deacetylation of the chitin polymer into chitosan by chitin deacetylases, chitosanases cleave chitosan into smaller subunits that can be further degraded, thereby providing carbon and nitrogen sources for other members of the microbiota (ref 14)."

Reviewer #2 (Remarks to the Author):

The authors explained all of my concerns related to the manuscript. I agree the publication at Nat. Comm.

Reviewers' Comments:

Reviewer #1:

Remarks to the Author:

The authors addressed all of my comments. I have no further suggestions. I believe this manuscript is ready for publication.